# Learning Physics Constrained Dynamics Using Autoencoders

**Tsung-Yen Yang**[1]    **Justinian Rosca**[2]    **Karthik Narasimhan**[1]    **Peter J. Ramadge**[1]
[1]Princeton University    [2]Siemens Corporation, Corporate Technology
yangtsungyen@gmail.com   justinian.rosca@siemens.com
{karthikn, ramadge}@princeton.edu

## Abstract

We consider the problem of estimating states (*e.g.,* position and velocity) and physical parameters (*e.g.,* friction, elasticity) from a sequence of observations when provided a dynamic equation that describes the behavior of the system. The dynamic equation can arise from first principles (*e.g.,* Newton's laws) and provide useful cues for learning, but its physical parameters are unknown. To address this problem, we propose a model that estimates states and physical parameters of the system using two main components. First, an autoencoder compresses a sequence of observations (*e.g.,* sensor measurements, pixel images) into a sequence for the state representation that is consistent with physics by including a simulation of the dynamic equation. Second, an estimator is coupled with the autoencoder to predict the values of the physical parameters. We also theoretically and empirically show that using Fourier feature mappings improves the generalization of the estimator in predicting physical parameters compared to raw state sequences when learning from high-frequency data. In our experiments on three visual and one sensor measurement tasks, our model imposes interpretability on latent states and achieves improved generalization performance for long-term prediction of system dynamics over state-of-the-art baselines.

## 1   Introduction

Neural networks have become a core computational component in domains such as computer vision [1], natural language processing [2], and deep reinforcement learning [3]. Recent work has shown that neural networks can exhibit an inductive bias that is often introduced via designing specific structures [4]. This bias can be used to encode prior task knowledge that helps the network generalize to unseen data. For example, convolution neural networks capture the translation invariance of key image features. In this spirit, we develop a structured neural network model that leverages a dynamic equation to estimate both the state of a dynamical system (*e.g.,* position and velocity) and its physical parameters (*e.g.,* friction constants) from a sequence of partial observations. Knowing the state and parameters of the system aids learning a control policy and tracking parameter value changes over time. For instance, for a self-driving car, we want to learn a neural network that estimates the vehicle's position and velocity from a sequence of egocentric camera images. The estimated state can then be used in a control policy. In addition, we want to track physical parameters over time, *e.g.,* friction coefficients. This is useful for vehicle maintenance and safety. We expect that exploiting a neural network, regularized to follow a dynamic equation, will streamline the required data-intensive operations and yield improved performance.

Observations can be direct measurements of some system variables (*e.g.,* accelerations), or take a more complex form (*e.g.,* images). We group such observations over specified time windows and also refer to these groupings as observations. We obtain a compact representation for these observations that permits observation reconstruction using an autoencoder (see Fig. 1(a)). $\{\boldsymbol{h}_s\}$ in Fig. 1(a) is

36th Conference on Neural Information Processing Systems (NeurIPS 2022).

a representation of an observation sequence $\{o_s\}$ over a specified time window, and $\{\hat{o}_s\}$ is the reconstructed observation sequence. Learning the autoencoder is both data and computationally intensive. In addition, $h_s$ may not be physically interpretable and be a "system state."

Motivated by these observations, we assume a simulator is provided that specifies the physical laws of the system. This model can arise from first principles (*e.g.,* Newton's laws), but its free parameters $\theta$ (*e.g.,* masses, lengths) remain to be specified. Given an initial state and $\theta$, the physics simulator generates a state trajectory $\{\hat{x}_s\}$ consistent with the laws of physics. To leverage this model, we require an estimator $f(\cdot)$ that maps a sequence of states $\{\tilde{x}_s\}$ to an estimate of $\theta$. We then couple the estimator $f$ and the physics simulator with the autoencoder as shown in Fig. 1(b). We train the autoencoder and $f(\cdot)$ to minimize the observation reconstruction loss $\sum_s \|o_s - \hat{o}_s\|_2^2$. Within this process, we train the encoder $h(\cdot)$ to minimize the sum of squared state errors: $\sum_s \|\tilde{x}_s - \hat{x}_s\|_2^2$. The complete model (Fig. 1(b)) is called *Autoencoder with Latent Physics* (ALPS).

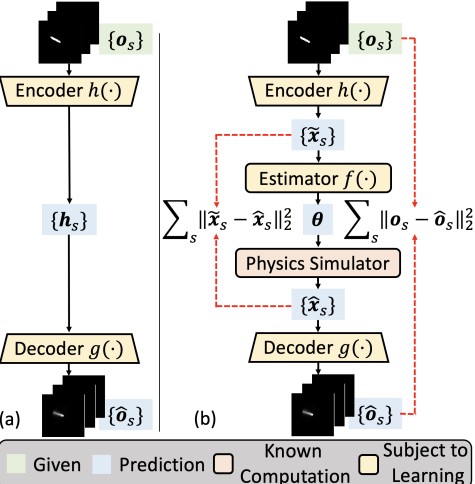

Figure 1: **(a)** An autoencoder learns a latent representation from a block of observations. **(b)** Combining a dynamic equation and parameter estimator in (a).

The paper's contributions are three-fold. **(1)** ALPS is the method that learns to identify the system parameters and the mappings between states and observations from data. In contrast, conventional system identification (*e.g.,* tools in Matlab) requires knowing the function mapping and identifying the parameters online, which is costly to run. **(2)** We show that one can learn periodic or vibrational behavior in this setting using a Fourier feature of states. **(3)** We evaluate ALPS in three simulated and one real-world train dynamics. ALPS can achieve up to 4.8x and 6.3x better physical parameter and state prediction accuracy, respectively, over prior approaches.

## 2   Related Work

**Physics-informed neural networks.** There is growing interest in including a physics prior or algebraic and logical constraints into neural networks [5, 6, 7, 8, 9, 10, 11, 12, 13, 14, 15, 16, 17, 18, 19, 20, 21, 22, 23, 24, 25, 26, 27, 28, 29]. For example, [30, 31, 32, 33, 34, 35, 36, 37, 38, 39, 40] exploit Lagrangian or Hamiltonian mechanics to learn an energy-conserving system based on position, momentum, and the derivatives thereof along trajectories. These works assume the physical parameters of the system are constant and need not be estimated. In contrast, we estimate the physical parameters. This is important for fault detection and localization, and for safety. Papers [41, 42, 43, 44] learn a general physical simulation from data, but their model is required to have *state* information or a *known* forward rendering engine to map states to observations. In contrast, we learn a physics-based autoencoder to estimate states in an unsupervised manner. Paper [45] also uses an autoencoder with physics to predict parameters. However, we show that learning state sequences as in [45] fails to generalize to unseen parameters when the system exhibits high-frequency behavior.

**Koopman-inspired neural networks.** Among many physics-informed neural networks developed in the past years, the recent effect has also considered applying deep learning in learning Koopman operators. We highlight several papers that are similar to our approach here. Koopman operators use an embedding to describe non-linear systems in a linear form. They are useful in analyzing the system dynamics but require domain knowledge to find the embedding. To overcome this challenge, the work [46, 47] use an autoencoder to identify Koopman eigenfunctions. In addition, the work [48, 49] also use an autoencoder for Koopman spectral analysis by learning Koopman invariant subspaces from data. The use of autoencoder structure is similar to our approach, but we explicitly use a physics simulator for constraining the representation of the autoencoder. Furthermore, the work [50] propose a method that optimizes neural networks by Koopman theory.

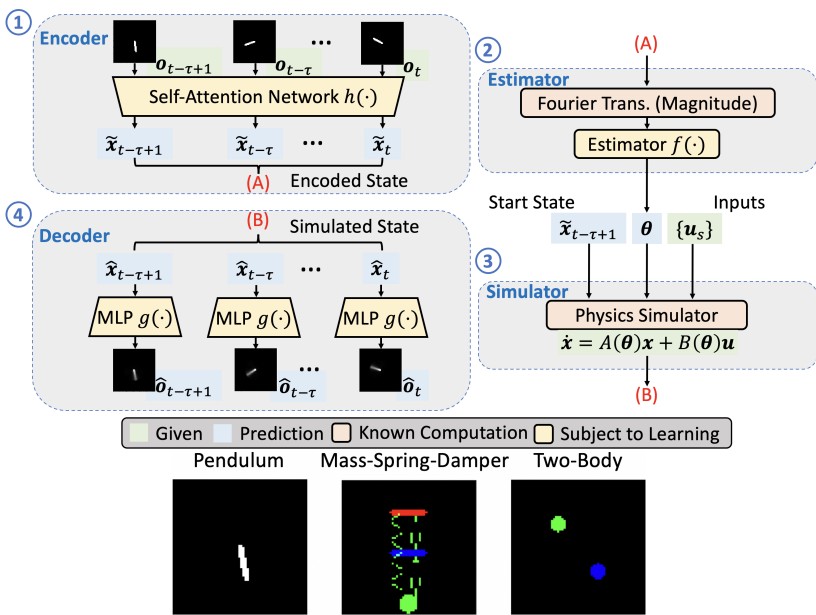

Figure 2: **ALPS and the tasks.** ALPS consists of four parts: **(1)** an encoder network that estimates states from observations, **(2)** a parameter estimator network that predicts physical parameters from a state sequence, **(3)** a physics simulator generates a state trajectory provided with an initial state and values for the physical parameters, and **(4)** a decoder network reconstructs observations from states.

**Fourier features for high-frequency data.** [51, 52] show that using Fourier features helps neural networks learn high-frequency content in image regression tasks. This approach requires specifying the Fourier series coefficients and the basis frequencies. In contrast, the Fourier features in our model are computed from the states, which allows us to predict physical parameters. In addition, the concurrent work [53] replaces the self-attention sublayers [54] with a Fourier transformation of the input word token in natural language processing. They show that this method is sufficient to capture semantic relationships in several text classification tasks. In this paper, we provide a theoretical justification for using Fourier features to learn system dynamics with high-frequency data.

**System identification.** The proposed approach uses the neural network to identify the system parameters from data, which is similar to conventional system identification methods such as linear grey-box model estimation (greyest) tool provided in Matlab [55]. However, ALPS does not require the mapping between the states and the observations, whereas the greyest method needs to provide full system dynamics. This offers an advantage when the observation space is in high dimension. In addition, it is possible to use Matlab model compilation to speed up the estimation of system parameters, which is similar to the analogy of training ALPS first and then deploying it during test time. However, even with Matlab model compilation, ALPS is still different since ALPS uses a neural network to identify the system parameters.

## 3 Problem Formulation

We consider the continuous-time system

$$\dot{\boldsymbol{x}} = \boldsymbol{A}\boldsymbol{x} + \boldsymbol{B}\boldsymbol{u}; \quad \boldsymbol{o} = g(\boldsymbol{x}), \tag{1}$$

where $\boldsymbol{A} \in \mathbb{R}^{n \times n}$, $\boldsymbol{B} \in \mathbb{R}^{n \times p}$ and $\boldsymbol{x} \in \mathbb{R}^n$, $\boldsymbol{u} \in \mathbb{R}^p$, $\boldsymbol{o} \in \mathbb{R}^q$, denote the state, input, and observation, respectively. The function $g$ provides a partial observation $\boldsymbol{o}$ of $\boldsymbol{x}$. In most situations we have partial knowledge of $\boldsymbol{A}$ and $\boldsymbol{B}$ from physics. This is true and reasonable in many large-scale industrial applications ranging from wind turbines to aircraft, where system designers use physics knowledge to design machines. Hence we assume mappings $A(\cdot) : \Theta \rightarrow \mathbb{R}^{n \times n}$ and $B(\cdot) : \Theta \rightarrow \mathbb{R}^{n \times p}$, from physical parameters $\boldsymbol{\theta} \in \Theta$ to the system matrices $A(\boldsymbol{\theta}), B(\boldsymbol{\theta})$, are given (*i.e.,* $\boldsymbol{A}$ and $\boldsymbol{B}$ obtain a specific sparse and parameteric form). In addition, Eq. (1) assumes the system dynamics to be linear, but our approach can be extended to other non-linear cases.

In practice, we only have observations at discrete points in time. For simplicity, we assume these are equally spaced at times $t = 0, 1, \ldots$. At sample time $t$, we have the window of observations $\{o_s\}_{s=t-\tau+1}^{t}$, where $\tau$ is the window length. We assume the sampling rate satisfies the Nyquist rate.

Our problem can now be stated as follows. Given functions $A(\cdot) : \Theta \to \mathbb{R}^{n \times n}$, $B(\cdot) : \Theta \to \mathbb{R}^{n \times p}$, we seek to learn a network that estimates a state sequence $\{x_s\}_{s=t-\tau+1}^{t}$, physical parameters $\theta$, and a mapping $g(\cdot)$ from a finite sequence of past observations $\{o_s\}_{s=t-\tau+1}^{t}$ and past known inputs $\{u_s\}_{s=t-\tau+1}^{t}$. Once trained, the network can predict states and has learned to adapt physical parameters to the context without re-training. Note that even though the network operates with time sampled variables, the physics simulator can be used to predict states at any time. This problem setup is distinct from that of HNN [33], HGN [36], or Symplectic ODE-Net [35]. In their setups, the model is only trained on data from a *single* physical parameter $\theta$, and hence they need to re-train networks for every new set of physical parameters. In addition, compared to conventional system identification approaches, we do not need to rerun the solver each time–we only need a forward pass of the network to get the estimation during deployment (testing time).

To make the learning problem well-defined, *i.e.,* to ensure that we can uniquely identify the unknown system parameters, we make the following assumption.

**Assumption 3.1. (Identifiability [56])**: For a sufficiently large $\tau$, the sequence of states and inputs $\{(x_s, u_s)\}_{s=t-\tau+1}^{t}$ uniquely specifies the unknown parameter $\theta$.

Parameter identifiability is a well-studied problem in system identification [56, 57, 58, 59, 60, 61]. For example, the (continuous time) system $\dot{x} = \begin{bmatrix} -(a+b) & b \\ b & -c \end{bmatrix} x + \begin{bmatrix} 1 \\ 0 \end{bmatrix} u$ with $o = \begin{bmatrix} 1 & 0 \end{bmatrix} x$, where $\theta = [a, b, c]^T$ and $g(\cdot)$ is a linear function, satisfies Assumption 3.1 when $b \neq 0$.

# 4 Nework Architecture

The network in Fig. 1(b) is expanded in Fig. 2 into its four parts: an encoder network, a parameter estimator network, a physics simulator, and a decoder network.

**The encoder network** $h(\cdot)$ **(from $\{o_s\}$ to $\{\tilde{x}_s\}$).** We consider two types of observations: **(1)** pixel images, or **(2)** direct measurements of some system variables. For the first case, we use a convolution neural network to compress a pixel observation $o$ into a compact vector embedding $z' \in \mathbb{R}^d$, which preserves image features. For the second case, we use a feedforward network to project an observation $o$ into some higher dimensional space with a vector embedding $z'$. In addition, to estimating states from these vector embeddings, we need to aggregate $\{z'_s\}$ to extract the local and global context of the dynamics. One approach is to use recurrent neural networks (RNN) (*e.g.,* Dreamer [62]). However, RNNs suffer from vanishing gradient problems and slow computation when processing long-term sequences. Hence we use a self-attention network [54] to attend to $z'$ of greatest importance to predict states and improve efficiency.

Furthermore, to inject a position signal of observations in the sequence, we add a positional encoding $p \in \mathbb{R}^d$ to $z'$ ($z := z' + p$), where $p$ are sine and cosine functions of different frequencies (see [54]). Finally, we stack embeddings over $\tau$ steps to form a matrix $Z \in \mathbb{R}^{\tau \times d}$.

The self-attention module can be formulated as querying a dictionary with key-value pairs associated with learnable weight matrices $W^Q \in \mathbb{R}^{d \times d_Q}$, $W^K \in \mathbb{R}^{d \times d_K}$, and $W^V \in \mathbb{R}^{d \times d_V}$ ($d_Q = d_K$ here): $\text{Attention}(Q, K, V) = \text{softmax}(\frac{QK^T}{\sqrt{d}})V$, where $Q := ZW^Q$, $K := ZW^K$, $V := ZW^V$, and the softmax is taken over the sequence length $\tau$. To provide multiview of the embedding, we use the multihead variant of the attention by concatenating each attention head along the sequence axis

$$\text{Multihead} := \text{Concat}(\text{head}_1, \ldots, \text{head}_i, \ldots, \text{head}_I)W^O, \quad \text{head}_i := \text{Attention}(Q, K, V),$$

where $W^O \in \mathbb{R}^{I d_V \times d}$ are learnable matrices, $I$ is the number of heads, and each attention head $i$ has its own learnable weight matrices $W_i^Q \in \mathbb{R}^{d \times d_Q}$, $W_i^K \in \mathbb{R}^{d \times d_K}$, and $W_i^V \in \mathbb{R}^{d \times d_V}$.

Finally, a feedforward network takes in the Multihead embedding (of size $\mathbb{R}^{\tau \times d}$) and produces the parameters of the distribution for each state in the sequence. The parameters are used to define a posterior distribution over the encoded state $\tilde{x}_s \sim Q(\cdot | \tilde{o}_s)$ with the prior $P(\tilde{x}_s)$. For a translational coordinate, the posterior distribution is a Gaussian distribution with a unit Gaussian prior. Hence the

network predicts a mean $\boldsymbol{\mu} \in \mathbb{R}^n$ and a standard deviation $\boldsymbol{\sigma} \in \mathbb{R}^n$ of a Gaussian distribution. In addition, for a rotational coordinate, the posterior distribution is a von Mises (vM) distribution with a unit vM prior. Similar to a Gaussian distribution, a vM distribution is defined by two parameters: a mean $\boldsymbol{\mu} \in \mathbb{R}^2$, $\|\boldsymbol{\mu}\|_2 = 1$ (the angular position $(\cos\varphi, \sin\varphi)$), and a concentration $\kappa \in \mathbb{R}^+$ around $\boldsymbol{\mu}$. Such parameterization is found useful in practice, *e.g.,* [63]. Appendix E provides details about self-attention networks for estimating states.

**The parameter estimator network** $f(\cdot)$ **(from $\{\tilde{\boldsymbol{x}}_s\}$ to $\boldsymbol{\theta}$).** The parameter estimator network predicts physical parameters from state sequences $\{\tilde{\boldsymbol{x}}_s\}$. However, for systems that involve periodic or vibrational behavior, prior work [51] has shown that neural networks fail to capture the high-frequency content in the data. To improve generalization and reduce the effect of noise, we do a Fourier transform on each component $j$ of state trajectories $\{\tilde{\boldsymbol{x}}_s(j)\}_{s=t-\tau+1}^t$ to get $\{\tilde{X}_\omega(j)\}_{\omega=t-\tau+1}^t$ : $\tilde{X}_\omega(j) := \sum_{k=t-\tau+1}^t \tilde{\boldsymbol{x}}_k(j)\Big[\cos\Big(\frac{2\pi}{\tau}\omega k\Big) - i \cdot \sin\Big(\frac{2\pi}{\tau}\omega k\Big)\Big]$. Note that for the systems that do not have periodic or vibrational behavior, using $\{\tilde{\boldsymbol{x}}_s(j)\}_{s=t-\tau+1}^t$ would be enough.

One may use $\{\tilde{X}_\omega(j)\}$ as features for the parameter estimator network to predict physical parameters. However in Section 5 we will show that by using the neural tangent kernel (NTK) theory [64], which treats neural networks as a kernel regression, the resulting kernel matrix of $\{\tilde{X}_\omega(j)\}$ does not preserve high-frequency components in the data. To solve this, we will show that using the *magnitude* of the Fourier features $\{|\tilde{X}_\omega(j)|\}$ alleviates the issue. Hence the parameter estimator network takes in a concatenation of $\{|\tilde{X}_\omega(j)|\}$ from each component of the state and predicts physical parameters $\boldsymbol{\theta}$.

**The physics simulator (from $\tilde{x}_{t-\tau+1}$ and $\boldsymbol{\theta}$ to $\{\hat{\boldsymbol{x}}_s\}$).** Given a start state $\tilde{x}_{t-\tau+1}$ (from the encoder's first state prediction) and values for the physical parameters $\boldsymbol{\theta}$, we use the neural ordinary differential equation (ODE) [65], a differential ODE solver, to generate a simulated state trajectory $\{\hat{\boldsymbol{x}}_s\}_{s=t-\tau+1}^t : \hat{\boldsymbol{x}}_{t-\tau+1}, \ldots, \hat{\boldsymbol{x}}_{t-1}, \hat{\boldsymbol{x}}_t = \text{ODESolver}(\tilde{x}_{t-\tau+1}, \dot{\boldsymbol{x}} = A(\boldsymbol{\theta})\boldsymbol{x} + B(\boldsymbol{\theta})\boldsymbol{u}, \tau, \Delta)$, where ODESolver takes in a start state, an ODE, a window length, and a sampling time interval $\Delta$. Note that $\hat{\boldsymbol{x}}_{t-\tau+1} = \tilde{x}_{t-\tau+1}$. In addition, using an ODE solver allows us to generate an accurate state trajectory compared to that of RNN as in [62].

**The decoder network** $g(\cdot)$ **(from $\hat{\boldsymbol{x}}_s$ to $\hat{\boldsymbol{o}}_s$).** Finally, the decoder network is either a deconvolutional network (for image observations) or a feedforward network (for sensor measurements) that takes in each individual ODE-simulated state $\hat{\boldsymbol{x}}_s$ and generates a reconstructed observation $\hat{\boldsymbol{o}}_s$.

**Discussion.** Note that we can replace the decoder $g$ with a differentiable rendering engine. Prior work [44] use a differentiable rendering engine to reconstruct the scene given the estimation of the system parameters and the states. However, using a differentiable rendering engine may introduce a simulation overhead, and using a network here is more generalizable. We think the future extension of ALPS to differentiable rendering engines is a valuable future research direction. In addition, one may think that the estimator network can directly predict the system parameters given the input observation without taking the states. The reason to predict the states is that for some applications, it may be useful to know the state of the system. For example, for a self-driving car, we would like to know the speed of other vehicles by using observations from cameras. Knowing the speed of other vehicles (*i.e.,* , the state) allows the self-driving car to plan for a trajectory, which is important for the safe deployment of the system. In addition, we would like to increase the interpretability of the model. The inclusion of the state allows the system designer to ensure the representation learned by neural networks is informative.

**The loss function.** Given a sequence of $\tau$ observations, we minimize the following loss function:

$$\mathcal{L} = \underbrace{\sum_{s=t-\tau+1}^t D_{\text{KL}}(Q(\tilde{\boldsymbol{x}}_s|\boldsymbol{o}_s)||P(\tilde{\boldsymbol{x}}_s))}_{\text{VAE loss for } h, f, \text{ and } g} + \underbrace{\sum_{s=t-\tau+1}^t \|\boldsymbol{o}_s - \hat{\boldsymbol{o}}_s\|_2^2}_{\text{Obs. recons. loss for } h, f, \text{ and } g} + \underbrace{\sum_{s=t-\tau+1}^t \|\tilde{\boldsymbol{x}}_s - \hat{\boldsymbol{x}}_s\|_2^2}_{\text{State recons. loss for } f \text{ and } h}.$$

The variational autoencoder (VAE) [66] loss is a variational bound on the marginal log-likelihood of the data. It is used to train the encoder $h$, the estimator $f$, and the decoder $g$. Using VAEs avoids learning degenerated solutions and provides stable training of the network over a deterministic network. In addition, the observation reconstruction loss encourages reconstructed observations $\{\hat{\boldsymbol{o}}_s\}$ to match true observations $\{\boldsymbol{o}_s\}$, and the state reconstruction loss constrains encoded states $\{\tilde{\boldsymbol{x}}_s\}$ to follow simulated states $\{\hat{\boldsymbol{x}}_s\}$ generated by physics. The former is used to train $h$, $f$, and $g$, and the latter is used to train $f$ and $h$. Both observation and state reconstruction losses are

important for training. We find that removing the state reconstruction term reduces the state prediction performance of the encoder since the network can predict arbitrary sequences. In addition, removing the observation reconstruction term impedes the image reconstruction quality, which is vital for training the whole model. In practice, we tune the weight for each loss term to accommodate different scales. Note that for the larger loss term, we use a weight smaller than 1 to ensure learning stability.

## 5 Fourier Feature Mappings for Learning Periodic System Dynamics

An important feature of ALPS is that it uses a Fourier feature mapping of states to learn periodic system dynamics. Prior works [67, 68] have empirically shown that using a Fourier feature can help with high frequency signals. To lay the foundation for the analysis and justify the use of Fourier features, in Section 5.1 we first review the recent work that uses NTK theory [64, 51]. This allows us to control the training of our parameter estimator network as fully-connected networks. Then in Section 5.2 we use these tools to analyze the effects of using Fourier features, their magnitudes, and their phases when predicting system parameters that involve the *periodic behavior*.

### 5.1 Deep networks as a kernel regression

Consider a set of labelled training data $\{(\boldsymbol{v}_i, y_i)\}_i^m$ with $\boldsymbol{v}_i \in \mathbb{R}^n$, $y_i \in \mathbb{R}$, and $i \in [1:m]$. Set $\boldsymbol{y} = [y_1, \ldots, y_m]^T \in \mathbb{R}^m$. Now bring in a feature map $\phi : \mathbb{R}^n \to \mathbb{R}^r$ with kernel $k(\boldsymbol{v}_i, \boldsymbol{v}_j) = \phi(\boldsymbol{v}_i)^T \phi(\boldsymbol{v}_j)$. Let $\boldsymbol{K} = [k(\boldsymbol{v}_i, \boldsymbol{v}_j)] \in \mathbb{R}^{m \times m}$ denote the kernel matrix for the training examples and $k(\boldsymbol{v}) = [k(\boldsymbol{v}_i, \boldsymbol{v})] \in \mathbb{R}^m$ denote the vector of kernel evaluations $k(\boldsymbol{v}_i, \boldsymbol{v}), i \in [1, m]$, for a test sample $\boldsymbol{v} \in \mathbb{R}^n$. The resulting kernel regression predictor is $\hat{y}(\boldsymbol{v}) = \boldsymbol{y}^T \boldsymbol{K}^{-1} k(\boldsymbol{v})$.

Now bring the concept of NTK proposed by [64]. The theory in [64] says that when the width of the layers of fully-connected deep networks with weights $\boldsymbol{w}$ initialized from a Gaussian distribution $\mathcal{N}$ tends to infinity, and the learning rate for stochastic gradient descent tends to zero, the neural network estimator $\hat{y}(\boldsymbol{v}; \boldsymbol{w})$ converges to the kernel regression solution using NTK. The NTK is defined as

$$k_{\text{NTK}}(\boldsymbol{v}_i, \boldsymbol{v}_j) = \mathbb{E}_{\boldsymbol{w} \sim \mathcal{N}} \left[ \left( \frac{\partial \hat{y}(\boldsymbol{v}_i; \boldsymbol{w})}{\partial \boldsymbol{w}} \right)^T \left( \frac{\partial \hat{y}(\boldsymbol{v}_j; \boldsymbol{w})}{\partial \boldsymbol{w}} \right) \right].$$

Under asymptotic conditions, a neural network's output after $t$ updates can be approximated as

$$\hat{y}^{(t)}(\boldsymbol{v}; \boldsymbol{w}) \approx \boldsymbol{y}^T (\boldsymbol{I} - e^{-\eta \boldsymbol{K} t}) \boldsymbol{K}^{-1} k(\boldsymbol{v}),$$

where $e^{\boldsymbol{M}}$ is the $n$ by $n$ matrix $\boldsymbol{M}$ given by the power series $e^{\boldsymbol{M}} = \sum_{i=0}^{\infty} \frac{1}{i!} \boldsymbol{M}^i$ with $\boldsymbol{M}^0 = \boldsymbol{I}$.

**Spectral bias in neural networks.** Now we want to compute the training error of a neural network after $t$ times update. Let $\boldsymbol{K} = \boldsymbol{U} \boldsymbol{\Sigma} \boldsymbol{U}^T$ denote the eigendecomposition of the kernel matrix $\boldsymbol{K}$ which must be positive semidefinite (PSD). Here $\boldsymbol{U}$ is an orthogonal matrix and $\boldsymbol{\Sigma}$ is a diagonal matrix whose entries are the nonnegative eigenvalues ordered by magnitude: $\lambda_1 \geq \lambda_2 \geq \cdots \geq \lambda_m \geq 0$. So the training error in terms of $L_2$ norm $\|\cdot\|_2^2$ is $\left\| \hat{\boldsymbol{y}}^{(t)} - \boldsymbol{y} \right\|_2^2 = \left\| \boldsymbol{U} \text{diag}([e^{-\eta \lambda_1 t}, \ldots, e^{-\eta \lambda_m t}]^T) \boldsymbol{U}^T \boldsymbol{y} \right\|_2^2$. This shows the training convergence will decay exponentially at the rate $\eta \lambda_i$. Hence the components in $\boldsymbol{y}$ will be learned faster if their corresponding eigenvalue is larger. In Section 5.2 we will show that for a sequence of states without doing Fourier transform, the resulting NTK will have smaller eigenvalues at the high-frequency components. This leads to a slower convergence in high-frequency components of data which are essential to identify parameters for the periodic behavior.

### 5.2 The effect of Fourier feature mapping

To understand the effect of the Fourier feature mapping, we first derive the kernel function of Fourier features, their magnitudes, and their phases. Then we compare the spatial bias of the kernel matrices from these three kernels.

**Kernels of Fourier feature mapping.** Consider a state trajectory $\boldsymbol{v} = [x_0, x_1, \ldots, x_{\tau-1}]^T$, where here we consider a 1D case for a scalar state $x \in \mathbb{R}$ although it can be extended to a vector state $\boldsymbol{x} \in \mathbb{R}^n$. The feature maps of the Fourier feature mapping, their magnitudes, and their phases are

**(1)** $\phi_{\text{DFT}}(\boldsymbol{v}) = [X_0, \ldots, X_\omega, \ldots, X_{\tau-1}]^T \in \mathbb{R}^\tau$;   **(2)** $\phi_{\text{MAG}}(\boldsymbol{v}) = [|X_0|, \ldots, |X_{\tau-1}|]^T \in \mathbb{R}^\tau$;

**(3)** $\phi_{\text{PHA}}(\boldsymbol{v}) = [\arg(X_0), \ldots, \arg(X_{\tau-1})]^T \in \mathbb{R}^\tau$,

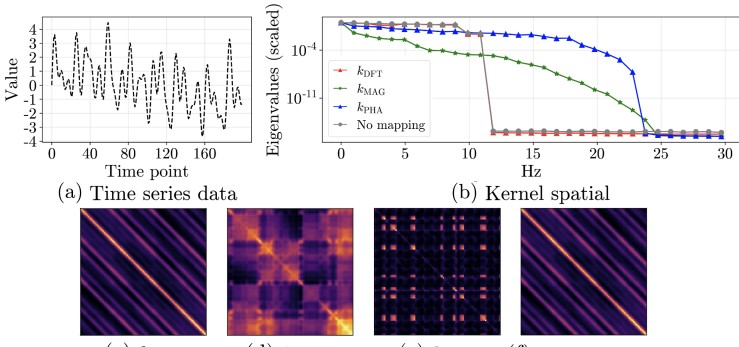

(a) Time series data      (b) Kernel spatial

(c) $k_{\mathrm{DFT}}$    (d) $k_{\mathrm{MAG}}$     (e) $k_{\mathrm{PHA}}$    (f) No mapping

Figure 3: Using a Fourier feature mapping of $k_{\mathrm{MAG}}$ and $k_{\mathrm{PHA}}$ results in a wider spectrum, which lets neural networks learn a wide frequency content. **(a)** The time series data with the sine basis frequency of $25, 17.5, 11, 7.7, 2, 1$Hz associated with the amplitude of $1, 1.2, 1, 1, 0.4, 1$. **(b)** The kernel spatial of the kernel matrices. We see that $k_{\mathrm{MAG}}$ can preserve high-frequency parts of the signals. **(c-f)** The kernel matrices of the composed NTK. (best viewed in color)

where $X_\omega = \sum_{j=0}^{\tau-1} x_j \left[ \cos\left(\frac{2\pi}{\tau}kj\right) - i \sin\left(\frac{2\pi}{\tau}kj\right)\right]$. Set $\boldsymbol{C}_k = \left[ \cos\left(\frac{2\pi}{\tau}ki\right) - \sin\left(\frac{2\pi}{\tau}kj\right)\right] \in \mathbb{R}^{\tau \times \tau}$, then the kernel functions of these mappings are

$$\textbf{(1)}\ k_{\mathrm{DFT}}(\boldsymbol{v}_1, \boldsymbol{v}_2) = \sum_{k=0}^{\tau-1} \boldsymbol{v}_1^T \boldsymbol{C}_k \boldsymbol{v}_2; \quad \textbf{(2)}\ k_{\mathrm{MAG}}(\boldsymbol{v}_1, \boldsymbol{v}_2) = \sum_{k=0}^{\tau-1} \sqrt{\boldsymbol{v}_1^T \boldsymbol{C}_k \boldsymbol{v}_1 \boldsymbol{v}_2^T \boldsymbol{C}_k \boldsymbol{v}_2};$$

$$\textbf{(3)}\ k_{\mathrm{PHA}}(\boldsymbol{v}_1, \boldsymbol{v}_2) = \phi_{\mathrm{PHA}}(\boldsymbol{v}_1)^T \phi_{\mathrm{PHA}}(\boldsymbol{v}_2).$$

After mapping the input points into the Fourier features, we feed them into a neural network to obtain $\hat{y}(\phi(\boldsymbol{v}); \boldsymbol{w})$. Hence for DFT kernel function the resulting composed kernel of the neural network is $k_{\mathrm{NTK}}(\phi_{\mathrm{DFT}}(\boldsymbol{v}_1), \phi_{\mathrm{DFT}}(\boldsymbol{v}_2))$. Similarly, we have the kernels for $k_{\mathrm{MAG}}$ and $k_{\mathrm{PHA}}$.

**Visualizing the composed NTK.** We generate time series data composed of different frequencies and magnitude of sine waves. The length of the data is 200 samples, and we set $\tau = 100$ (*i.e.,* $\boldsymbol{v} \in \mathbb{R}^{100}$) with the sampling rate being 100Hz. We slide a window and hence have 101 instances in total. Fig. 3 shows the time series data, the effects of each kernel, and its spatial plot. By construction, $k_{\mathrm{MAG}}$ and $k_{\mathrm{PHA}}$ have a slower decay in the high-frequency domain as shown in Fig. 3(b). In addition, $k_{\mathrm{DFT}}$ and no mapping have the same kernel matrix and a narrower kernel spatial. This observation supports the idea of not using $X_\omega$ when learning the system with periodic or vibrational behavior as $|X_\omega|$ (magnitude) preserves high-frequency information, which is useful for parameter estimations. Note that for the system that does not have high-frequency signals, both $X_\omega$ and $|X_\omega|$ are equally well.

**Comparison to [51].** Our work is inspired by [51], which also uses Fourier feature mappings. However, there are a few key differences. **(1) Setup.** The Fourier feature mapping in [51] transforms the low-dimensional x-y coordinates into high-dimensional Fourier features, which *projects* data into a high-dimensional space. In contrast, we use Fourier feature mapping of raw time series data, which *compresses* data into more compact representations. **(2) Analysis**. We discuss the difference of Fourier feature mappings, their magnitudes, and their phases to understand the effect of each mapping for predicting system parameters. In contrast, [51] does not have such analysis.

# 6   Simulations

We study the following questions: **(1)** How does ALPS perform compared to other baseline methods without the Fourier feature mapping and physics-in-the-loop? **(2)** What is the effect of self-attention networks in ALPS? **(3)** How does ALPS perform in a full-scale train wheel system dynamics?

**Visual dataset.** We generate three visual datasets: pendulum, mass-spring-damper (MSD), and a two-body system to compare the performance of ALPS to that of the baseline in the literature. These baselines are very commonly used in the prior work. We first randomly sample an initial state and physical parameters, and then generate a 125 step rollout following the true system dynamics, and render corresponding 64 by 64 by 3 pixel observation snapshots. The sampling rate is 20Hz in the

pendulum, 100Hz in MSD, and 6Hz in two-body systems. The observation length $\tau$ is 100. In total, we generate 500 training and 500 test trajectories, resulting in 13,000 training and test sequences.

**(1) Pendulum.** The dynamics is $\frac{d}{dt}\begin{bmatrix} \varphi \\ \dot{\varphi} \end{bmatrix} = \begin{bmatrix} \dot{\varphi} \\ -\beta\dot{\varphi} - \frac{G}{L}\sin(\varphi) \end{bmatrix}$, where $\varphi$ is an angle, $\dot{\varphi}$ is an angular velocity, $\beta$ is a friction coefficient, $G$ is a gravitational constant, and $L$ is the length of the pendulum. We fix $G = 10$, $L = 1$, then sample $\beta$ from a uniform distribution $\beta \sim \mathbb{U}(0.1, 1)$, an initial angle from a uniform distribution $\varphi_0 \sim \mathbb{U}(-\pi, +\pi)$, and an initial angular velocity from a uniform distribution $\dot{\varphi}_0 \sim \mathbb{U}(0.5, 4)$. We predict $\boldsymbol{\theta} = [\beta]$ in this task.

**(2) Mass-spring-damper.** The dynamics of double mass-spring-damper system is

$$\frac{d}{dt}\begin{bmatrix} x(1) \\ x(2) \\ x(3) \\ x(4) \end{bmatrix} = \begin{bmatrix} 0 & 1 & 0 & 0 \\ \frac{-\alpha(1)-\alpha(2)}{m(1)} & \frac{-\alpha(3)-\alpha(4)}{m(1)} & \frac{\alpha(1)}{m(1)} & \frac{\alpha(3)}{m(1)} \\ 0 & 0 & 0 & 1 \\ \frac{\alpha(1)}{m(2)} & \frac{\alpha(3)}{m(2)} & -\frac{\alpha(1)}{m(2)} & \frac{\alpha(3)}{m(2)} \end{bmatrix} \begin{bmatrix} x(1) \\ x(2) \\ x(3) \\ x(4) \end{bmatrix} + \begin{bmatrix} 0 & 0 \\ \frac{\alpha(2)}{m(1)} & \frac{\alpha(4)}{m(1)} \\ 0 & 0 \\ 0 & 0 \end{bmatrix} \begin{bmatrix} u(1) \\ u(2) \end{bmatrix},$$

where $x(1), x(3)$ are the displacement and velocity of the primary spring; $x(2), x(4)$ are the displacement and velocity of the secondary spring; $u(1), u(2)$ are the inputs; $\alpha(2), \alpha(4)$ are the stiffness and damping ratio of the primary spring and damper; $\alpha(1), \alpha(3)$ are the stiffness and damping ratio of the secondary spring and damper, and $m(1), m(2)$ are the mass of the primary and secondary spring. We fix all the parameters and the initial state except for $\alpha(2)$, which is sampled from a uniform distribution $\alpha(2) \sim \mathbb{U}(4 \times 10^3, 4 \times 10^6)$. The input excitations are sampled from a unit Gaussian distribution $\boldsymbol{u} \sim \mathcal{N}(0, 0.05) \times \mathcal{N}(0, 0.05)$ (*i.e.,* white noise). We predict $\boldsymbol{\theta} = [\alpha(2)]$ in this task.

**(3) Two-body.** In this system two particles interact with each other via an attractive force. The dynamics is a Hamiltonian $\mathcal{H} = \frac{\|p(1)\|_2^2}{2m(1)} + \frac{\|p(2)\|_2^2}{2m(2)} + \frac{Gm(1)m(2)}{\|q(1)-q(2)\|_2}$, where $p(1), p(2)$ are the positions of the particles; $q(1), q(2)$ are the momentum of the particles; $m(1), m(2)$ are the masses of the particles, and $G$ is a gravitational constant. We fix $G = 10$, $m(1) = m(2) = 1$, then sample $\mathcal{H}$ from a uniform distribution $\mathcal{H} \sim \mathbb{U}(0.4, 1)$. We predict $\boldsymbol{\theta} = [\mathcal{H}]$ in this task.

**Time series dataset.** In addition, we obtain an MSD system dataset with state measurements.

**(4) MSD time series data from full-scale train wheel suspension system.** To show the applicability of ALPS, we conduct experiments with the data that represent the situation when sensors are installed in the real train wheel suspension system. We use full-scale dynamics of the train wheel suspension system with 18 state elements, 12 observations, 16 input excitations, and 24 system parameters. We want to identify the stiffness and damping ratio of the 12 spring-dampers. The dynamics are complex due to the interactions of multiple springs and dampers. In addition, this problem is challenging as the data contain noise. To align with the real world scenario, we assume that the system parameters of the system slowly change over time. As a result, the data has the same set of system parameters but with different initial conditions (*e.g.,* initial vibration speed). The dataset contains 20 trajectories with 500 steps sampled by 100Hz. We use a 50-50 split to get the training and test datasets with $\tau = 100$. Here we *do not* use the self-attention and decoder network in ALPS–the estimator takes in state measurements directly.

**Baselines.** We select the following baselines that are commonly used in the literature.

**(1) Context-aware dynamics model (CDM) [69].** CDM is the state-of-the-art method to learn the system dynamics into two stages: it first learns a context vector that captures the local dynamics, and then predicts the next state based on the context vector and the current state and input. CDM *does not* consider physics prior *nor* Fourier feature mapping. This is to show that using physics priors improves performance and benchmark the results.
**(2) Autoencoder.** We remove the estimator and the physics simulator to benchmark the mismatch between the true state and the latent representation (equal dimension) of the autoencoder in Fig. 1(a) to show the interpretability of ALPS.
**(3) ALPS w/o the Fourier feature mapping.** We consider a variant by replacing the Fourier feature mapping with raw encoded state trajectories $\{\tilde{\boldsymbol{x}}_s\}$. This method is *similar* to [45], which also uses time series data to learn system dynamics.
**(4) ALPS w/o self-attention networks.** We consider a variant by replacing the self-attention network with a simple MLP to predict the position of the system, followed by a first-order finite-difference estimator to estimate the velocity. This uses the same approach as in [63] to estimate states. Finally,

| | Pendulum | | | Mass-Spring-Damper | | | Two-body | | |
|---|---|---|---|---|---|---|---|---|---|
| | SE | OE | PE | SE | OE | PE | SE | OE | PE |
| CDM [69] | 345.22 | 1766.70 | – | 0.99 | **5.69×10⁸** | – | 50.23 | **2.03×10⁸** | – |
| Autoencoder | 3041.12 | **600.84** | – | 7.63 | 7.42×10⁸ | – | 95.80 | 2.71×10⁸ | – |
| **ALPS (ours)** | **86.48** | 1696.91 | **0.06** | **0.29** | 7.43×10⁸ | **0.60×10⁶** | **0.45** | 2.59×10⁸ | **0.02** |
| w/o Fourier feat. [45] | 90.82 | 1773.39 | 0.29 | 0.93 | 7.44×10⁸ | 2.28×10⁶ | 1.86 | 2.56×10⁸ | 0.02 |
| w/o self-attention [63] | 181.22 | 1950.77 | **0.06** | 1.85 | 7.44×10⁸ | 1.87×10⁶ | 511.49 | 2.72×10⁸ | 0.27 |

Figure 4: **(Top)** Evaluation for the tested networks in the visual tasks. **(Bottom)** Selected visualization results (we randomly sampled the ground truths and tested ALPS and CDM to show more cases). Note that for the pendulum and two-body tasks, the range of $\boldsymbol{\theta}$ is $(0, 10]$; for the MSD tasks, the $\boldsymbol{\theta}$ is in the range of $10^6$. SE: state prediction error; OE: observation prediction error; PE: parameter prediction error. And in visualization the true state here is $\boldsymbol{x}$, ENC state is $\tilde{\boldsymbol{x}}$, and ODE state is $\hat{\boldsymbol{x}}$. ALPS achieves competitive performance in predicting physical parameters and states.

we ensure that all baselines are comparable in terms of representation power and the number of parameters. The loss functions used for these baselines are included in Appendix D.

**Evaluation metrics.** We use a mean squared error (MSE) between simulated $\{\hat{\boldsymbol{x}}_s\}$ and true states $\{\boldsymbol{x}_s\}$ to evaluate the performance: $\text{SE} := \frac{1}{N} \sum_{i=1}^{N} \sum_{s=t-\tau+1}^{t} \|\hat{\boldsymbol{x}}_s - \boldsymbol{x}_s\|_2^2$, where $N$ is the number of test data. We also compute MSE between reconstructed $\{\hat{\boldsymbol{o}}_t\}$ and true observations $\{\boldsymbol{o}_t\}$ : $\text{OE} := \frac{1}{N} \sum_{i=1}^{N} \sum_{s=t-\tau+1}^{t} \|\hat{\boldsymbol{o}}_s - \boldsymbol{o}_s\|_2^2$. Moreover, we compute the absolute value difference between estimated $\hat{\boldsymbol{\theta}}$ and true parameters $\boldsymbol{\theta}$ : $\text{PE} := \frac{1}{N} \sum_{i=1}^{N} \left\| \hat{\boldsymbol{\theta}} - \boldsymbol{\theta} \right\|_1$.

**Results in the visual tasks.** Fig. 4 shows the results. Note that we let the size of the latent representation of the autoencoder be the same as the size of the true states. The reason is that we would like to know its state error when there is no constraint imposed so that we can quantify the difference from the one with imposing physics constraints. We see that **(1)** ALPS achieves the best performance in predicting physical parameters in all cases, with at the most 4.8x lower error in the pendulum task. On the other hand, the Fourier feature mapping does not have much effect in predicting the physical parameter in the two-body system. This verifies the analysis of the Fourier feature mapping: for the task with a wider frequency spectrum (*e.g.,* 100Hz in the MSD), the Fourier feature improves the prediction due to higher eigenvalues in the high-frequency content, whereas for the task with only a low-frequency spectrum (*e.g.,* 6Hz in the two-body), raw state sequences can already capture low-frequency content. This supports the idea of *using Fourier feature mappings for high-frequency data.* **(2)** ALPS achieves competitive results in predicting the states, with at the most 6.3x lower error in the MSD task. We further visualize the state prediction results in the pudendum task for ALPS, which shows that ALPS can accurately track the true states. Without self-attention networks, the network has a substantial SE due to a large error in computing the velocity, as in the two-body task. The smaller the sampling rate is, the greater the velocity estimation error is. In addition, the high SE in autoencoder suggests that its latent representation is uninterpretable. This verifies the idea of using physics to constrain the representation of the autoencoder to estimate states.

In addition, **(3)** ALPS achieves comparable results in reconstructing observations. The low error for CDM in the MSD and two-body tasks is due to degenerated solutions as shown in the Fig. 4. This implies that using physics stabilizes the training and improves the reconstruction of observations. **(4)** Finally, the autoencoder baseline has higher reconstruction loss in some tasks. This is because

that it learns a degenerated solution, producing blur images due to high-frequency movements of objects on the scene. Fig. 8 in Appendix E provides more qualitative visualizations of reconstructed images. Specifically, we find that CDM fails to track the state and resulting blur images in the mass-spring-damper system or duplicated objects in the two-body system. In contrast, ALPS can precisely track the state and reconstruct the image well. Our remaining simulations explore ALPS's ability in predicting physical parameters from raw state measurements.

**Results in MSD time series data from full-scale train wheel suspension system.** To scale the proposed approach to a more complex system, we apply the approach in a system with a full-scale train wheel suspension system. The system contains 18 state elements, 12 observations, 16 input excitations, and 24 system parameters. Fig. 5 shows the

|  | SE | PE |
|---|---|---|
| CDM [69] | 56.04 | – |
| **ALPS (ours)** | **3.02** | **0.44×10$^4$** |

Figure 5: Results in the MSD system for predicting physical parameters from time series data. Our method achieves the best performance.

results. Here ALPS does not use the encoder nor decoder network–the estimator takes in state measurements directly, as the system is fully-observable in the dataset. Overall we see that **(1)** ALPS achieves the best performance. **(2)** CDM has worse SE since an MLP cannot learn well from data with high-frequency content. In addition, we find that the average prediction error rate (*i.e.,* $\frac{\hat{\theta}-\theta}{\theta}$) for these 24 parameters is *0.42%* on the complex system. This also implies that by providing physics to the network, we can improve SE. These observations show ALPS can robustly identify parameters simultaneously from state measurements, and support the theory of using Fourier features to learn periodic dynamics. Moreover, this result shows ALPS can be deployed in real prognostic applications for tracking physical parameters to enhance railroad safety. Our result is significant for system designers to diagnose the system and hence schedule maintenance.

## 7 Conclusion

We addressed the problem of predicting states and physical parameters of a system from observations with dynamic equations. We showed that the latent representation of the autoencoder is uninterpretable. We then use dynamic equations to constrain the latent representation of the autoencoder to be consistent with the laws of physics. We analyzed the effect of Fourier features for estimating physical parameters. The results showed that ALPS achieves competitive performance in the visual tasks and the time series dataset with up to 24 parameter predictions at the same time. This shows that our method scales to high-dimensional systems and it is useful for real-world applications.

**Limitation and future work.** Future work could improve ALPS in several ways. For instance, the understanding of underlying mechanisms in self-attention networks for estimating states from observations can be advanced. One path is to probe the network by using approaches in natural language processing. In addition, we require to have dynamic equations and assume that the system is linear and exhibits periodic or vibrational behaviors. These make it challenging to generalize to more complicated settings such as contact dynamics. One solution is to combine known physics with neural models (*e.g.,* interaction networks [13]) to compensate for modeling error, and use time-domain and frequency-domain features together to predict physical parameters. Moreover, it would be interesting to explore other more complex tasks (*e.g.,* non-linear systems, fluid dynamics, molecular simulation). Another limitation of ALPS is the overhead caused by simulating trajectories. For the training of ALPS, the computation cost mostly comes from running the differential solver. For example, under the same batch size of 10 and one gradient update, on average ALPS requires 498 seconds whereas the autoencoder requires 317 seconds, which is 57% slower. The training time for CDM is also similar to the autoencoder. A method to overcome this will be valuable. In addition to the challenge of computation cost, the further investigation of the full system with higher dimensional system parameters (*e.g.,* large-scale machines with a hundred system parameters) and noise observation is an important future research direction. Finally, incorporating the ideas from the literature on the system identification [59, 60, 61] to ALPS will further enhance the performance of ALPS.

## Acknowledgments and Disclosure of Funding

The authors would like to thank the anonymous reviewers and the area chair for their insightful and helpful comments and feedback. Tsung-Yen Yang thanks Siemens Corporation, Corporate Technology for their support.

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
