# OpenReview forum: "Learning Physics Constrained Dynamics Using Autoencoders"
_NeurIPS.cc/2022/Conference — NeurIPS 2022 Accept_

### Official Review · Reviewer_dT3B · 2022-07-06

**Rating:** 3
**Confidence:** 4
**Soundness:** 2 fair
**Presentation:** 2 fair
**Contribution:** 2 fair

**Summary:**

This is an interesting paper. It addresses the task of predicting states and physical parameters of a system from observations using dynamic equations and autoencoder methods.

**Questions:**

1. Why not compare your approach with standard system identification methods from engineering: MATLAB will do this automatically for the types of examples you use.
2. equation 1: is the system time-invariant (LTI)?
3. line 171ff: what assumptions do you make about the knowledge of physics?


**Limitations:**

A better empirical analysis is needed. You must fix the fundamental mismatch between the input data (images) and the empirical validation (wheel suspension system, Mass-spring-damper, pendulum).

Is this approach to be deployed on-line? Will the training be done off-line only?


line 71, section 2 "Related Work" is highly limited. There is a huge literature on learning dynamical systems that is much more relevant than PINNs.
 --System identification is one area (you reference [50-52] later.
 --Nagel and Huber[2021] "Autoencoder Identification of LTI systems", ECC-2021.

Also see:

@article{markovsky2021behavioral,
  title={Behavioral systems theory in data-driven analysis, signal processing, and control},
  author={Markovsky, Ivan and D{\"o}rfler, Florian},
  journal={Annual Reviews in Control},
  volume={52},
  pages={42--64},
  year={2021},
  publisher={Elsevier}
}
@article{padoan2022behavioral,
  title={Behavioral uncertainty quantification for data-driven control},
  author={Padoan, Alberto and Coulson, Jeremy and van Waarde, Henk J and Lygeros, John and D{\"o}rfler, Florian},
  journal={arXiv preprint arXiv:2204.02671},
  year={2022}
}


**Strengths And Weaknesses:**

Overall, it is difficult to understand what has been achieved. First, the empirical analysis does not provide clear evidence that the proposed approach improves quite basic engineering system identification methods, as provided in tools like MATLAB. The examples are trivial, and the firepower of the proposed approach needs more complex examples to show itself. Second, it is unclear what assumptions you make about the knowledge of physics in the basic design; the empirical analysis has very strong physics models, so reading just the empirical analysis seems to say that this is little different to LTI system identification methods.

Other Issues
============
One main issue that I have is the fundamental mismatch between the input data (images) and the empirical validation (wheel suspension system, Mass-spring-damper, pendulum). In real-world systems of this kind, typically the data will be from cheap sensors, e.g., accelerometer, position, etc.). This type of time-series input is easier to deal with than image data, and so the approach should be more successful.

Is this meant to work with systems like autonomous vehicles, where image data is integrated with lidar, accelerometer, etc. data? What is the target application domain? It is extremely unlikely that a car manufacturer would install a camera to monitor a wheel suspension system, due to cost above all.

It would be helpful to have a diagram of your full pipeline--it's hard to piece this together from the fragmented descriptions of pieces of inference.

A second fundamental issue is that the Simulations, section 6, only show system identification examples, with the help of autoencoders. The pendulum example has a well-defined ODE, and just needs to identify angle and friction coefficient. In fact, you only predict friction coefficient.
Analogous issues are with the other 2 examples. For mass-spring damper, the physics is pretty comprehensive, and in fact all you estimate is \theta = [a(2)].





 Other issues:

line 41: claim "not guaranteed to have the Markov property, i.e., be a system state."----why is this true? Do you require ONLY Markov models?


Your approach is significantly more complex than a basic autoencoder. Please address the relative computation complexity wrt a standard baseline, and describe the additional data and time (training, online) as compared to the baseline.
Obviously, there will be a tradeoff between relative computation complexity and accuracy, and this should be reported.

Figure 5:a)  image is blurred since it is proprietary. Please delete it as it adds zero content.

---

> ### Author Response · Authors · 2022-08-01
> **Response**
>
> We appreciate the time spent by the reviewer on this draft and thank the reviewer for the helpful comments. Before we answer the questions, we would like to emphasize (a) the assumption, (b) the setup, and (c) the contribution of ALPS here.
>
> **Assumption:** In the paper, we assume that the system dynamics are linear and the system parameters are fixed. However, the mapping from the state to the observation need not be linear (as stated in Line 95). Moreover, ALPS can also work for the non-linear system such as the pendulum task, as we showed in the paper. Second, we assume that system dynamics (Eq. (1)) are given, but the system parameters and mapping between the states and observations are unknown. As a result, ALPS is required to learn the encoder and decoder in order to identify the states and system parameters, which is different from tools in MATLAB. The solver in MATLAB requires the mappings between states and observations. These are two major assumptions that make ALPS different from MATLAB tools.
>
> **Setup:** It is true that the pendulum task, two-body task, and mass-spring-damper task can be solved using “grey-box system identification tools (greyest)” from MATLAB. However, when the observation space is high-dimensional, it is expensive to run MATLAB “online” when the data arrives. Instead, we would like to first train a neural network with data collected from the past, and then deploy the neural network (no further training) to predict the system parameters directly without simulation online. Hence, for setup in MATLAB, there is no notion of training and test time, whereas, in our setup, we first train ALPS and then deploy it in the run-time system. As a result, ALPS can run faster than MATLAB tools during deployment (i.e., test time) since ALPS only needs a single forward pass to estimate the parameters.
>
> **Contribution:** We would like to emphasize the contribution of the ALPS is a framework that can estimate states and system parameters from observations using differential simulators. Compared to tools like MATLAB, we use the training-test scheme of neural networks to reduce the overhead of using MATLAB online. In addition, the second contribution is that when the mapping from the observation to the state is unknown, ALPS can learn this mapping from data, whereas MATLAB requires to specify the mapping. For example, it is useful to estimate the speeds (i.e., states) of the surrounding vehicles from camera images (i.e., observations) for self-driving cars. Finally, we provide a theoretical analysis for using Fourier features, which is important to understand how neural networks learn to identify parameters.
>
> We make a table here to compare the difference between ALPS and MATLAB for fast reference:
> |   |      Ours      |  Tools from MATLAB **(without model compilation)** |
> |----------|:-------------:|:-------------:|
> | Assumption |  Known dynamics, but not given mappings between states and observations | Known dynamics, and mappings between states and observations |
> | Setup |     Train networks first, and then deploy the network   |   Deploy the solver online|
> | Pros and Cons | (1) Fast deployment during runtime (2) Learn to predict system parameters without the mapping between states and observations  |   (1) Overhead during run-time (2) Require to know the mapping between states and observations |
>
> As pointed out by the reviewer, it is possible to use MATLAB model compilation to speed up the estimation of system parameters. However, even with MATLAB model compilation, ALPS is still different since ALPS uses a neural network to identify the system parameters; whereas MATLAB model compilation does not. Using neural networks can potentially scale up more complicated systems. We will add the discussion in the paper.

---

> > ### Author Response · Authors · 2022-08-01
> > **Response (Cont'd)**
> >
> > Now, we answer the questions below.
> >
> > *1. Overall, it is difficult to understand what has been achieved. First, the empirical analysis does not provide clear evidence that the proposed approach improves quite basic engineering system identification methods, as provided in tools like MATLAB. Second, it is unclear what assumptions you make about the knowledge of physics in the basic design; The empirical analysis has very strong physics models, so reading just the empirical analysis seems to say that this is little different to LTI system identification methods.*
> >
> > **Ans:** As we pointed out in the beginning, tools like MATLAB require a mapping from the states to observations; whereas ALPS learns this mapping directly from data. ALPS only needs to know the system dynamics equation (i.e., $x’=Ax+Bu$). In addition, once we train ALPS offline, we can deploy ALPS online with a single forward pass to estimate the system parameters, whereas tools like MATLAB require running the solver online. Hence, our contribution is a new deep learning-based approach that uses a different setup and assumptions than MATLAB. Please see the table for comparison, and please let us know if there are further questions.
> >
> > *2. Is this meant to work with systems like autonomous vehicles, where image data is integrated with lidar, accelerometer, etc. data? What is the target application domain?*
> >
> > **Ans:** Yes, you are correct. For example, it is useful to estimate the speeds (i.e., states) of the surrounding vehicles from camera images (i.e., observations) for self-driving cars. Such an approach is important for self-driving cars that do not have lidar sensors such as Tesla.
> >
> > *3. A second fundamental issue is that the Simulations, section 6, only show system identification examples, with the help of autoencoders. The pendulum example has a well-defined ODE, and just needs to identify angle and friction coefficient. In fact, you only predict friction coefficient. Analogous issues are with the other 2 examples. For mass-spring damper, the physics is pretty comprehensive, and in fact all you estimate is \theta = [a(2)].*
> >
> > **Ans:** In the original draft, we also estimated all the 24 system parameters of real-world MSD systems based on the state. During the rebuttal, we also tested ALPS given the input of the observation. In both cases, ALPS achieves an average accuracy of 0.41% and 1.61%, respectively. As a result, ALPS is able to scale up from 1 or 2 parameters to 24 parameters. The inclusion of the pendulum and other simple tasks allows us to understand and analyze the ALPS. In the future, we would like to test ALPS in more challenging tasks such as fluid simulation.
> >
> > *4. It would be helpful to have a diagram of your full pipeline*
> >
> > **Ans:** In the paper, we started by outlining the main difference between the vanilla autoencoder and the one with an estimation of states constrained by the physics shown in Fig. 1. Then in Fig. 2, we provided a diagram of our full pipeline using the visual pendulum task as an example. Please let us know if there is anything unclear to the reviewer in Fig. 1 and Fig. 2. Thank you!
> >
> > *5. Not guaranteed to have the Markov property, i.e., be a system state."----why is this true? Do you require ONLY Markov models?*
> >
> > **Ans:** Our intention is just to say that the latent representation learned by the autoencoder without physics need not be the true system state. In other words, our goal is to constrain this latent representation to be the true state. We have removed this in the revised paper to avoid confusion. Thank you!

---

> > > ### Author Response · Authors · 2022-08-01
> > > **Response (Cont'd)**
> > >
> > > *6. Your approach is significantly more complex than a basic autoencoder. Please address the relative computation complexity wrt a standard baseline, and describe the additional data and time (training, online) as compared to the baseline. Obviously, there will be a tradeoff between relative computation complexity and accuracy, and this should be reported.*
> > >
> > > **Ans:** We are sorry for not reporting this. Yes, you are correct. For the training of ALPS, the computation cost mostly comes from running the differential solver. For example, under the condition of the same batch size of 50, one gradient update, and visual mass-spring-damper task, on average ALPS requires 498 seconds whereas the autoencoder requires 317 seconds for training, which is 57% of the increase (Intel i9 CPU Macbook Pro). The training time for CDM is also similar to the autoencoder. Although there is a computation cost, ALPS can achieve better accuracy during runtime deployment (test time).
> > >
> > > In addition, during testing, we found that ALPS estimates the system parameters in less than 0.5 seconds. In contrast, the tools from MATLAB take about 20 seconds due to transformation from the states to the image observations, which is 40 times slower than ALPS. We have added the discussion in the revised paper. We will include more stats in the appendix. The future improvement could lie in developing a faster solver for ALPS to learn efficiently.
> > >
> > > *7. Figure 5:a) image is blurred since it is proprietary. Please delete it as it adds zero content.*
> > >
> > > **Ans:** Yes, you are correct. We have removed the figure in the revised paper.
> > >
> > > *8. Why not compare your approach with standard system identification methods from engineering: MATLAB will do this automatically for the types of examples you use.*
> > >
> > > **Ans:** As we stated in question 1, the setup and the assumption are very different. MATLAB tools do not have a notion of training and test, and hence the algorithm runs online. In addition, it requires knowing the mapping from the state to the observation. In contrast, ALPS first trains a neural network and then deploys online. At the same time, it learns the mapping from the data. These two factors prevent us from using MATLAB tools. Please see the discussion in the beginning. We have updated the contribution in the introduction section to make it clear in the revised paper.
> > >
> > > *9. equation 1: is the system time-invariant (LTI)?*
> > >
> > > **Ans:** Yes, the system could be time-invariant. However, we want to emphasize that the setting and assumption are different from the system identification literature. We want to train neural networks to estimate the system parameters and the function mapping need not be provided.
> > >
> > > *10. line 171ff: what assumptions do you make about the knowledge of physics?*
> > >
> > > **Ans:** If we understand correctly, you are referring to line 171. If it is the case, then the assumption is that as long as we have a differential physics simulator, we can use ALPS since we can take a gradient to update the parameter of ALPS (the decoder, estimator, and encoder).
> > >
> > > *11. You must fix the fundamental mismatch between the input data (images) and the empirical validation (wheel suspension system, Mass-spring-damper, pendulum).*
> > >
> > > **Ans:** Yes, you are correct. In some applications, we have sensors to measure states or observations of the system. As a result, in the original draft, we also tested the setting where ALPS is used to identify 24 system parameters from real-world sensor data (see the paragraph “Results in real MSD time series data”). We found that ALPS works well in this setting. In addition, as we discussed in question 2, we are also interested in the setting where the sensor measurement is not provided but camera images are provided. For example, this is useful for the self-driving car to estimate the speed of other vehicles from camera images. Please let us know if there are any other questions. Thank you!
> > >
> > > *12. Is this approach to be deployed on-line? Will the training be done off-line only?*
> > >
> > > **Ans:** Yes, you are correct. This approach is to be deployed online and the training is done offline. Please read the paragraph in the beginning.
> > >
> > > *13. Missing citations*
> > >
> > > **Ans:** Thank you so much for pointing these papers out. We have added them to the revised paper.
> > >
> > > **Finally, we thank the reviewer for providing valuable insight and helpful comments from a system identification perspective. We acknowledge that the field of system identification has been working on this problem, and the tools developed are important for the academic and industry. We think ALPS provides another perspective to use neural networks to identify system parameters. Combining ALPS with the approach in system identification is a valuable future research direction.**

---

> > ### Comment · Reviewer_dT3B · 2022-08-08
> > **model compilation**
> >
> > Thanks to the authors for the detailed comment. I would like to clarify that ALPS is actually closer to MATLAB in model compilation than the authors understand. Just as one can train a neural net and used the "compiled" model online, one can train the parameters of a MATLAB model and use the compiled model online as well, with significant speedups. So in the table provided in the rebuttal the "setup" and "Pros/Cons" sections are actually not that different.

---

> > > ### Author Response · Authors · 2022-08-08
> > > **Need clarification**
> > >
> > > Dear Reviewer,
> > >
> > > Thank you so much for the comments!
> > >
> > > Could you please clarify what you mean by *"one can train the parameters of a MATLAB model and use the compiled model online as well, with significant speedups"*? If we understand correctly, do you mean identifying the system parameters based on the data to get the initial estimates, and then during the deployment, you run the solver again with the parameter initialized with these initial estimates so that you can speed up the identification process? If this is the case, then we respectfully disagree that this approach is similar to ALPS. Here are the main differences:
> > >
> > > **(1)** We would like to reiterate the setup is different. See the above table.
> > >
> > > **(2)** What if you have a high-dimensional observation space, and the true system dynamics are very different from the initial estimates of the solver? Then the solver requires more time to identify the system parameters since the initial estimates are very off from the true values.
> > >
> > > **(3)** We have run the simulation and please see explanation 6. We showed that the MATLAB solver could be 40 times slower than ALPS in that case.
> > >
> > > Finally, we acknowledge much effort has been done into the system identification literature, and we have cited the relevant work in the revised paper including the papers you mentioned. Our work follows the line of work that combines a neural network with physics such as Greydanus et al., Hamiltonian neural networks, NeurIPS 2019, and Murthy et al., gradSim: Differentiable simulation for system identification and visuomotor control, ICLR, 2021. This work has received attention from the community. In addition, our work is also useful for the work in the model-based RL, in which the researchers aim to use a neural network to learn the transition function. With the help of physics, ALPS can provide a better estimate of the future state. Furthermore, we have addressed all the other comments, and please let us know if you have more concerns. Alternatively, if you feel that your original concerns are addressed, we would appreciate updating your evaluation to reflect that.

---

> > > > ### Comment · Reviewer_dT3B · 2022-08-09
> > > > **compilation of models**
> > > >
> > > > Here is the requested clarification: one can "compile" a neural net by training the model and creating a standalone executable for decision making. Similarly, the MATLAB Simulink Compiler enables you to compile Simulink simulations as standalone executables that do not require one to  run the solver again. Of course there are differences between the two approaches, but this to me is a clear analog in the use of compiler-based methods for generating decision-making models. And both approaches yield significant speedups in compiled form.
> > > >
> > > > I will of course update my review, but to me there are still significant overlaps with  the system identification literature that need to incorporated into your article. I still see that your provided table is inaccurate given the clarification.
> > > >
> > > > Regarding "(3) We have run the simulation and please see explanation 6. We showed that the MATLAB solver could be 40 times slower than ALPS in that case", I think that you may have done an unfair comparison given you did not understand this compiled model approach.
> > > >
> > > > I view comment (2) as irrelevant, since *every* model trained where the true system dynamics are very different from the initial estimates of the solver will be inaccurate, ALPS included.

---

> > > > > ### Author Response · Authors · 2022-08-09
> > > > > **Matlab Simulink Compiler**
> > > > >
> > > > > Dear Reviewer,
> > > > >
> > > > > Thank you so much for your clarification!
> > > > >
> > > > > We checked the MATLAB tool via the following resources from MathWorks:
> > > > >
> > > > > (1) Video about how Simulink works
> > > > > https://www.mathworks.com/products/simulink-compiler.html
> > > > >
> > > > > (2) Tutorial video about standalone executables  https://www.mathworks.com/videos/deploy-simulink-standalone-web-apps-integrate-enterprise-system-1599589985922.html
> > > > >
> > > > > (3) User interface for running system identification
> > > > > https://www.mathworks.com/videos/introduction-to-system-identification-toolbox-68901.html
> > > > >
> > > > > Based on these documents, we agree that compiling Simulink simulations as standalone executables could streamline the deployment process. However, as you mentioned, there are differences between the two approaches such as how the model exactly estimates system parameters. **We expect that even with the model compilation, the MATLAB tool still runs a gradient type of algorithm to identify the system parameters, as there is an optimization step shows up in the interface (please see the above videos)** Our approach uses neural networks with a single forward pass to estimate the parameters.
> > > > >
> > > > > In addition, based on the argument of compiling neural networks and analog of MATLAB compilation of models, *every* paper about deep learning could be seen as a concept of system identification, in which we try to *identify* the function mapping $f$ from the input data $X$ to output label $Y$. However, researchers in the field still see the benefit of applying neural networks in multiple areas since the pioneering paper Learning representations by back-propagating errors, 1986, by Rumelhart, David E; Hinton, Geoffrey E; Williams, Ronald J. As a result, our paper does not diminish the value of work from system identification. In fact, it provides another approach to solving the problem.
> > > > >
> > > > > Regarding our points (2) and (3), we will add the comparison and discussion to MATLAB Simulink Compiler in the paper. We will also update the table of the comparison.
> > > > >
> > > > > Once again, we thank the reviewer for discussing this with us!

---

> ### Author Response · Authors · 2022-08-08
> **Thank you again for your review.**
>
> Dear Reviewer,
> Thank you so much for your review -- please let us know if you have any remaining questions or concerns so that we can address them before the deadline tomorrow. Alternatively, if you feel that your original concerns are addressed, we would appreciate updating your evaluation to reflect that. Thank you!

---

### Official Review · Reviewer_2274 · 2022-07-11

**Rating:** 7
**Confidence:** 2
**Soundness:** 3 good
**Presentation:** 4 excellent
**Contribution:** 3 good

**Summary:**

This paper proposes a method for estimating states of a physical system along with the system parameters. It can be seen as a VAE encoder-decoder set up where the decoder comprises a physics simulator (neural ODE solver) so that the latents (physical states and parameters) are forced to be interpretable. Important features of the encoder that let this method work well are 1. self-attention for estimating states from observations and 2. Fourier features for estimating parameters from states. Experiments confirm that these feature are crucial. The method is successfully applied to a real-world problem.

**Questions:**

What are the limitations of the physics simulator. Does it need to be representable as a differential equation (possibly non-linear)? What about a simulator written as a program? It seems like the only requirement is differentiability with respect to the initial state and the physical parameters.

**Strengths And Weaknesses:**

This paper tackles an important problem of learning how to estimate interpretable physical states and parameters from observations. It's clearly written. The experiments are well-selected and empirically verify the important components of the system (Fourier features and self-attention). Moreover, it is successfully applied to a real-world problem.

I didn't fully grasp the significance of applying the NTK theory.

I'm giving this paper a 7 (Accept) based on my understanding but I'm not an expert.

---

> ### Author Response · Authors · 2022-08-01
> **Response**
>
> We appreciate the time spent by the reviewer on this draft and thank the reviewer for the helpful comments. We answer the questions below.
>
> *1. What are the limitations of the physics simulator. Does it need to be representable as a differential equation (possibly non-linear)? What about a simulator written as a program? It seems like the only requirement is differentiability with respect to the initial state and the physical parameters.*
>
> **Ans:** Yes, you are correct. As long as the physics simulator is differentiable, we can use it in ALPS. One possible limitation of the physics simulator is that during training, it may simulate slowly and hence the network would coverage slowly. This can be mitigated by using fast CPUs and GPUs to accelerate simulation. We have added the discussion about the limitations of the physics simulator in the revised paper.
>
> *2. I didn't fully grasp the significance of applying the NTK theory.*
>
> **Ans:** The NTK theory allows us to understand what kind of representation is critical for predicting the system parameters from observations. In the paper, we analyzed four representations: raw data, Fourier feature, phase of the Fourier feature, and magnitude of the Fourier feature. We found that by using the NTK theory, only the magnitude of the Fourier feature has a good representation to predict the system parameters. The analysis provides a foundation for the ALPS to use such a representation. Please let us know if there are additional questions. Thank you!

---

### Official Review · Reviewer_qaG1 · 2022-07-11

**Rating:** 7
**Confidence:** 4
**Soundness:** 3 good
**Presentation:** 3 good
**Contribution:** 3 good

**Summary:**

This manuscript proposes ALPS, Autoencoder with Latent Physics, which uses neural networks as the feature extractor and parameter regressor. A differentaible physics simulator is used to assign physical meanings to the latent space so that it follows physical laws. The manuscript compare their method on several diverse tasks including a complex real-world example.

**Questions:**

A major question I would raise is the difference between $\theta$ and $x$. Since $\theta$ can be used as the most accurate descriptor of the physical dynamics (because it supports physical simulation), it makes more sense to use the network to directly regress $\theta$ and skip $x$. Is there a reason behind it?

**Limitations:**

See weaknesses.

**Strengths And Weaknesses:**

The overall presentation is very clear to me. The method makes sense to me and the pipeline is clear. There are motivating examples and also advanced examples. After training on each environment, this parameter estimation network can be deployed without re-training or re-running any costly solver.

I would not call them weaknesses but I would like to know more about some choices the authors made in the manuscript. In my opinion, decoder $g(\cdot)$ can be replaced with a differentiable rendering and save the effort of learning a visualization system. I am aware that using a network here might be more generalizable to other scenarios which are hard to model, but I would like to know if it is a possibility.

---

> ### Author Response · Authors · 2022-08-01
> **Response**
>
> We appreciate the time spent by the reviewer on this draft and thank the reviewer for the helpful comments. We answer the questions below.
>
> *1. I would not call them weaknesses but I would like to know more about some choices the authors made in the manuscript. In my opinion, decoder g can be replaced with a differentiable rendering and save the effort of learning a visualization system. I am aware that using a network here might be more generalizable to other scenarios which are hard to model, but I would like to know if it is a possibility.*
>
> **Ans:** Thank you for pointing this out. Yes, we can replace the decoder $g$ with a differentiable rendering engine. Prior work by Murthy et al., ICLR 2021 uses a differentiable rendering engine to reconstruct the scene given the estimation of the system parameters. However, using a differentiable rendering engine may introduce a simulation overhead, and using a network here is more generalizable as you mentioned. We think the future extension of ALPS to differentiable rendering engines is a valuable future research direction.
>
> Murthy et al., gradSim: Differentiable simulation for system identification and visuomotor control, ICLR, 2021
>
> *2. A major question I would raise is the difference between θ and x. Since θ can be used as the most accurate descriptor of the physical dynamics (because it supports physical simulation), it makes more sense to use the network to directly regress θ and skip x. Is there a reason behind it?*
>
> **Ans:** The reason to predict the states is that for some applications, it may be useful to know the state of the system. For example, for a self-driving car, we would like to know the speed of other vehicles by using observations from cameras. Knowing the speed of other vehicles (i.e., the state) allows the self-driving car to plan for a trajectory, which is important for safe deployment of the system.
>
> In addition, removing the state information may affect the performance of the estimator. This is because it is unclear if the representation produced by the encoder is meaningful for the estimation of the system parameters.
>
> Moreover, we would like to increase the interpretability of the model. Inclusion of the state allows the system designer to ensure the representation learned by neural networks is informative.
>
> As a result, ALPS is a general model that can predict system parameters and states of the system for better performance and interpretability.

---

> > ### Comment · Reviewer_qaG1 · 2022-08-08
> > **Response to authors**
> >
> > Thank you for answering the questions. They sound reasonable to me, and I would like to keep the score.

---

> > > ### Author Response · Authors · 2022-08-08
> > > **Thank you!**
> > >
> > > Dear Reviewer,
> > > Thank you so much for the response!

---

### Official Review · Reviewer_Eodc · 2022-07-11

**Rating:** 6
**Confidence:** 4
**Soundness:** 3 good
**Presentation:** 3 good
**Contribution:** 3 good

**Summary:**

This paper proposed an autoencoder-based model for system identification (estimating unknown parameters of a system defined by a dynamic equation). The model is shown to improve predicting long-term system dynamics with unknown physical parameters by predicting both the physical parameters and the system state.

**Questions:**

How is state error (SE) measured in the autoencoder baseline when it only reconstructs the observations without predicting states?

Fig 4 Two-Body Observation Reconstruction: why are the GTs for ALPS and CDM different?

Weighting on loss function: it’s mentioned that `In practice, we find using the same weight for each loss term works well. However, in Fig. 4, it’s shown that SE, OE, and PE are of very different magnitude. If the same loss weight is used for each term, OE would dominate the loss for ALPS.

Scale differences in Fig 4 and Fig 6:
E.g. For Two-body, the loss converged to ~4,000 in fig 6 while SE+OE ~= 2.59*10^8 in Figure 4, and loss = VAE loss + SE +OE according to Section 4


**Limitations:**

The authors are upfront about the following two limitations of the work:

It assumes prior knowledge of the system,e.g. dynamics equations.

It only works for linear systems with periodic or vibrational behaviors.

One additional limitation the reviewer would like to point out:
As mentioned in Weaknesses, it lacks experimental results of the full system with higher dimensional system parameters. Such results would make the paper much stronger for real system applications.


**Strengths And Weaknesses:**

### Strengths
While incorporating physics into Deep RL is not novel, this paper proposes to explicit predict the physical parameter which has shown promising results for predicting system dynamics

Provided theoretical and empirical evidence for the benefit of Fourier Mapping of system states with a high-frequency component. As shown in the experiments, Fourier features boost the performance by a large margin, especially for State Error and Parameter Error.

### Weaknesses
Experiments for the full system (including the decoder) only include systems with one-dimensional system parameters. Experiments with 24 system parameters take the state measurement directly.

Code was included in the supplemental material. It is modified from other codebases however all READMEs are from the original code base and outdated. Thus it’s impossible to reproduce the results with the code in its current state.

---

> ### Author Response · Authors · 2022-08-01
> **Response**
>
> We appreciate the time spent by the reviewer on this draft and thank the reviewer for the helpful comments. We answer the questions below.
>
> *1. Experiments for the full system (including the decoder) only include systems with one-dimensional system parameters. Experiments with 24 system parameters take the state measurement directly.*
>
> **Ans:** In the dataset of 24 system parameters, both states and observations are included. To reduce the complexity, ALPS takes in the state measurement directly. During the rebuttal, we also ran an experiment with ALPS taking in the observations. We found that under the same setup as in Line 334, ALPS is able to estimate the system parameters with an average accuracy of 1.61%, which is slightly above 0.42% when using full states. This shows that ALPS can still work well for observations. We have included the discussion in the revised paper.
>
> *2. Code was included in the supplemental material. It is modified from other codebases however all READMEs are from the original code base and outdated. Thus it’s impossible to reproduce the results with the code in its current state.*
>
> **Ans:** We thank the reviewer for pointing this out. We will release the code once the paper has been polished.
>
> *3. How is state error (SE) measured in the autoencoder baseline when it only reconstructs the observations without predicting states?*
>
> **Ans:** We let the size of the latent representation of the autoencoder be the same as the size of the true states. The reason is that we would like to know its state error when there is no constraint imposed so that we can quantify the difference from the one with imposing physics constraints.
>
> *4. Fig 4 Two-Body Observation Reconstruction: why are the GTs for ALPS and CDM different?*
>
> **Ans:** We randomly sampled the ground truths and then tested ALPS and CDM to show more cases. We do not do cherry-picking. We have made this clear in the revised paper.
>
> *5. Weighting on loss function: it’s mentioned that `In practice, we find using the same weight for each loss term works well. However, in Fig. 4, it’s shown that SE, OE, and PE are of very different magnitude. If the same loss weight is used for each term, OE would dominate the loss for ALPS.*
>
> **Ans:** When we compute the loss function in the visual pendulum task, we indeed used the same weight. However, for the other tasks such as the visual mass-spring-damper system, we used different weights for each term. We are sorry for the typos. We have revised the paper.
>
> *6. Scale differences in Fig 4 and Fig 6: E.g. For Two-body, the loss converged to ~4,000 in fig 6 while SE+OE ~= 2.59*10^*8* *in Figure 4, and loss = VAE loss + SE +OE according to Section 4*
>
> **Ans:** You are correct. For Fig. 4, we reported the original SE and OE. For Fig 6, we used the weights for OE and SE that are smaller than 1 when the OE and SE terms are large. We have revised the paper.
>
> *7. One additional limitation the reviewer would like to point out: As mentioned in Weaknesses, it lacks experimental results of the full system with higher dimensional system parameters. Such results would make the paper much stronger for real system applications.*
>
> **Ans:** We thank the reviewer for pointing this out. We carefully verified the performance of ALPS by gradually increasing the dimension of system parameters: from one system parameter in the visual pendulum task to the 24 system parameters in the real MSD task. We agree that further investigation of the full system with higher dimensional system parameters (e.g., large-scale machines with a hundred system parameters) is an important future research direction.

---

### Author Response · Authors · 2022-08-01
**General Response**

We thank the reviewer for the helpful comments. We appreciate (1) the reviewer Eodc for finding our theoretical and empirical results convincing; (2) the reviewer qaG1 for identifying the usefulness of ALPS and finding our presentation is clear; (3) the reviewer 2274 for stating our paper is well-written and experiments are well-selected; and (4) the reviewer dT3B for providing helpful discussion. We have updated the paper based on the comments. Please see our response below and let us know if there is something missing. Thank you.

---

> ### Author Response · Authors · 2022-08-03
> **Please kindly let us know if you have any questions, thank you!**
>
> Please kindly let us know if you have any questions, thank you!

---

> ### Author Response · Authors · 2022-08-04
> **Please kindly let us know if you have any questions, thank you!**
>
> Please kindly let us know if you have any questions, thank you!  Much appreciate!

---

> ### Author Response · Authors · 2022-08-07
> **Please kindly let us know if you have any questions, thank you!**
>
> Please kindly let us know if you have any questions, thank you! Much appreciate!

---

### Author Response · Authors · 2022-08-09
**Final Check-In**

Dear Reviewers,

We would like to thank everyone for the helpful discussion and comments over the past few days. Please let us know if you have any pending questions!

Good luck with everything!

Best,

Authors

---

### Meta-Review · Area_Chair_n4ia · 2022-08-26

**Recommendation:** Accept
**Confidence:** Less certain

**Metareview:**

The paper builds a deep learning-based framework for estimating states and physical parameters by embedding a differential physics simulator into an autoencoder architecture. Integrating physics into neural networks is an interesting research area, and this paper proposes some interesting and novel ideas such as explicitly predicting the physical parameter of a given problem from data. The reviewers acknowledged the relevance of the proposed method and generally appreciated the results. The paper is nicely written, but the experiments are somewhat limited. Also the related work section is very shallow, ignoring to discuss recent work on Koopman-inspired autoencoders for scientific problems that has appeared at ICLR, ICML and NeurIPS in recent years. This said, I want to thank the authors for their detailed responses that helped in answering some of the reviewers' questions. The reviewers have provided detailed feedback in their reviews, and we strongly encourage the authors to incorporate this feedback when preparing a revised version of the paper. I particularly stress this, because the authors have not sufficiently incorporated the feedback of previous reviews at ICML. (In general, I feel that this is bad practice!! In this particular case I will have an additional look at the camera ready version of this paper.) Nevertheless, the overall feedback of the reviewers is positive (reviewer dT3B noted that he will raise his score), and I feel that this paper has the potential to motivate future research in this area. Thus, I am leaning toward suggesting a (weak) accept of this paper.

**Award:**

No

---

### Decision · Program_Chairs · 2022-09-14

Accept